# "Because of COVID. . .": The impacts of COVID-19 on First Nation people accessing the HIV cascade of care in Manitoba, Canada

Linda Larcombe[1,2,3]*, Laurie Ringaert[4], Gayle Restall[5], Albert McLeod[6], Elizabeth Hydesmith[7], Ann Favel[1], Melissa Morris[1], Michael Payne[8], Rusty Souleymanov[9], Yoav Keynan[1,2,3], Kelly MacDonald[1], Matthew Singer[8], Jared Star[9], Pamela Orr[1,2,3]

1 Department of Internal Medicine, Max Rady College of Medicine, University of Manitoba, Winnipeg, MB, Canada, 2 Department of Medical Microbiology and Infectious Diseases, Max Rady College of Medicine, University of Manitoba, Winnipeg, MB, Canada, 3 Department of Community Health Sciences, Max Rady College of Medicine, University of Manitoba, Winnipeg, MB, Canada, 4 Change Weavers Consulting, Winnipeg, MB, Canada, 5 Department of Occupational Therapy, Max Rady College of Medicine, University of Manitoba, Winnipeg, MB, Canada, 6 2Spirit Consultants, Winnipeg, MB, Canada, 7 Department of Anthropology, University of Manitoba, Winnipeg, MB, Canada, 8 Nine Circles Community Health Centre, Winnipeg, MB, Canada, 9 Faculty of Social Work, University of Manitoba, Winnipeg, MB, Canada

* Linda.larcombe@umanitoba.ca

**Data Availability Statement:** Relevant data has been provided in the paper. The complete raw minimal data cannot be publicly shared because it

## Abstract

### Background

The COVID-19 pandemic (March 2020-May 2023) had a profound effect around the world with vulnerable people being particularly affected, including worsening existing health inequalities. This article explores the impact of the pandemic on health services for First Nations people living with HIV (FN-PWLE) in Manitoba, Canada. This study investigated perceptions of both health care providers and FN-PWLE through qualitative interviews occurring between July 2020 and February 2022 to understand their experience and identify lessons learned that could be translated into health system changes.

### Methods

Using a qualitative, participatory-action, intentional decolonizing approach for this study we included an Indigenous knowledge keeper and Indigenous research associates with lived experience as part of the study team. A total of twenty-five [25] in-depth semi-structured interviews **were conducted** with eleven healthcare providers (HCPs) and fourteen First Nation people with lived HIV experience (FN-PWLE). In total, 18/25 or 72% of the study participants self-identified as First Nation people.

### Results

The COVID-19 pandemic negatively impacted health services access for FN-PWLE, a) disrupted relationships between FN-PWLE and healthcare providers, b) disrupted access to testing, in-person appointments, and medications, and c) intersectional stigma was compounded. Though, the COVID-19 pandemic also led to positive effects, including the creation of innovative solutions for the health system overall.

contains potentially identifying or sensitive participant information. The University of Manitoba Research Ethics Board has stipulated that all identifiable data be stored in a secure fileserver and cannot be shared publicly. Data are available upon request from The University of Manitoba Research Ethics Board via email (bannreb@umanitoba.ca) for researchers who meet the criteria for access to confidential data.

**Funding:** Canadian Institutes of Health Research (CIHR) - Mapping the Journey: Developing Culturally Appropriate, Geographically-Responsive HIV Care for Northern Manitoba First Nation People. CIHR Operating Grant (Grant Number CBA-164022). The funders had no role in study design, data collection and analysis, the decision to publish, or the preparation of the manuscript.

**Competing interests:** The authors declare no conflicts or competing interests.

## Conclusions

The COVID-19 pandemic exaggerated pre-existing barriers and facilitators for Manitoba FN-PWLE accessing and using the healthcare system. COVID-19 impacted health system facilitators such as relationships and supports, particularly for First Nation people who are structurally disadvantaged and needing more wrap-around care to address social determinants of health. Innovations during times of crisis, included novel ways to improve access to care and medications, illustrated how the health system can quickly provide solutions to long-standing barriers, especially for geographical barriers. Lessons learned from the COVID-19 pandemic should be considered for improvements to the health system's HIV cascade of care for FN-PWLE and other health system improvements for First Nations people with other chronic diseases and conditions. Finally, this study illustrates the value of qualitative and First Nation decolonizing research methods. Further studies are needed, working together with First Nations organizations and communities, to apply these recommendations and innovations to change health care and people's lives.

## Introduction

This study examines how the COVID-19 pandemic public health orders impacted the care for First Nations people living with HIV in the Canadian province of Manitoba. Health inequities experienced by marginalized groups were intensified by the global COVID-19 pandemic, including among those living with, and those at risk of HIV infection, as well as among minority groups, including Indigenous peoples in various countries [1–5]. Indigenous peoples in Canada (First Nation, Inuit and Métis) have a long history of suffering due to historical and more recent pandemics and epidemics, including tuberculosis and influenza [6]. Although biological determinants have been identified in some cases [7, 8], the primary causative factors for the disproportionate burden of infectious disease morbidity and mortality borne by Canadian Indigenous peoples are socioeconomic and political in nature [9, 10]. In 2019, before the advent of the COVID-19 pandemic, there were 121 new cases of HIV reported in Manitoba–a 13.1% increase compared to 2018 [11]. However, when the COVID-19 pandemic occurred in the province, laboratory testing for HIV decreased by 1.7% between 2019 and 2020 [5, 11]. During that time, the provincial government attributed the reduction in non-COVID-19 related healthcare services "to reduced access to care during months with the highest COVID-19 restrictions and fear of attending healthcare settings due to COVID-19" [5].

Few studies have looked at the impact of COVID-19 on First Nation peoples, and this is one of the few focused on First Nation peoples living with HIV in Manitoba [12, 13]. Our study documents experiential narratives of First Nations people living with HIV infection during the COVID-19 pandemic. We anticipated that hearing about the "healthcare journeys" of First Nation people living with HIV during the pandemic might provide ideas and guidance for improving the HIV health care.

## Methods

### Study design

We used a qualitative, participatory-action, intentional decolonizing approach for this study. Decolonizing methodology is a process through which non-First Nations research partners

commit to thinking, feeling, understanding and creating actions in solidarity (allyship) with First Nation people and communities [14]. This decolonizing work and allyship are based on the values of justice, equity, caring and respect. Our decolonizing approach to the research involved three aspects: (1) Our team included a First Nation Knowledge Keeper who guided the research design and was part of the data analysis process, (2) Our study included a decolonizing approach, using a two-eyed seeing lens [15]. The research team integrate First Nation and Western views into the design, implementation, analysis (sense-making) and recommendations for health system changes. In addition, we focused on building and creating ethical space through a series of workshops that our study First Nation Knowledge Keeper led with the entire study team [14].

Another key component of our study design involved hiring and training two First Nation peer research associates (people with lived First Nation and HIV experience) who assisted with the project development, interviews and data analysis. The peer research associates were trained in qualitative interview methods, ethics, and data analysis, and were supported throughout the project through open discussions and reflection [16]. They each brought their knowledge, experience and insights, which are unique to people with lived experience, to the interviews and analysis [17, 18]. This collaborative approach to data analysis provided alternative perspectives for creating themes relevant to First Nations people with lived experience [19]. Involving these research associates in the planning, interviews, and analysis, as well as the review of this paper, were important and intentional to challenge the perceptions of academic researchers on the study team [20, 21]. By embracing our team's diversity, we incorporated common guiding principles of First Nations research: collaboration, relationships, interconnectedness, connection to community, and respect for diverse forms of knowledge and lived experience [22].

## Ethics

This study was approved by the University of Manitoba Health Ethics Review Board (HS23123) and the Manitoba First Nation Health and Social Secretariat's (FNHSM's) Health Information Research Governance Committee (HIRGC). HCPs and FN-PWLE were enrolled into the study with written informed consent. "Healthcare provider" (HCP) was broadly defined for this study and included any front-line workers who were working with people with HIV, including family physicians, specialists, nurses, public health personnel, social workers, educators, community-based organization workers, program planners and others.

## Recruitment of study participants

The study was advertised via posters and shared through word of mouth by healthcare providers and First Nation people living with HIV residing in Manitoba. The posters were faxed to thirty-one nursing stations and health centers in Manitoba, posted on social media and posted in clinics and health centers in Winnipeg, the southern capital of Manitoba. The enrollment period for this study phase was approximately 1.5 years between July 2020 and February 2022.

Interested participants contacted the research team via telephone or email and were screened for eligibility. Inclusion criteria for participants who had lived experience of HIV included self-identification as a First Nation person, residence in Manitoba and age 18 years of age or older. HCPs had to have had experience with First Nation people living with HIV in Manitoba, either with direct care or through HIV programs that directly serve First Nation people living with HIV. Each participant was assigned a study number for the downstream anonymous analysis of the data.

## Interview process

Interviews took place between July 2020 and February 2022. This length of time for interviews was needed due to the various restrictions that were occurring during the COVID-19 pandemic. HCP interviews were conducted until June 2021. Two study team members interviewed each HCP participant using Zoom video conferencing. The research associates were undergoing training during the time that the HCPs were being interviewed, therefore, they did not participate. FN-PWLE were enrolled in the study beginning in April 2021 and ending in February 2022. Each FN-PWLE participant was interviewed by two study team members, including a research associate with lived HIV experience on Zoom either with or without the video feature.

Data collection involved conducting interviews with both FN-PWLE and HCPs between July 2020 and February 2022. Interviews with HCPs occurred between July 2021 and June 2022, and did not include research associates due to them being engaged in research training. Interviews with FN-PWLE occurred between April 2021 and February 2022, and were conducted with two study team members and one research associate with lived experience of HIV.

Demographic information (sex, gender (self-defined descriptor including heterosexual, two-spirit, bisexual, gay, asexual), ethnicity (non-Indigenous, First Nation (Dene, Cree, Ojibwa, Oji-Cree) and age were also collected from each participant. In addition, all participants were asked if, and when, they had had an HIV test.

All participating FN-PWLE and HCPs were asked the same series of semi-structured questions relating to their experience or understanding of facilitators and barriers for accessing healthcare at each of the five milestones that make up the HIV cascade of care: milestone 1-healthcare prior to positive HIV test; milestone 2- HIV positive test; milestone 3—getting linked to care; milestone 4—starting and being on medication; milestone 5–6 being on antiretrovirals and living with HIV. At each milestone, participants were asked to describe if, and how, COVID-19 had changed the process or their experience of HIV healthcare.

Our interviews followed a trauma-informed and decolonizing approach [23, 24]. All participants were offered support (emotional/mental health, social services) during and at the conclusion of the interviews, were offered breaks during the interviews, and were given the option of ending or changing the direction of the interview. We ensured they had control over their interview experience to mitigate traumatic responses and create space for the emotions that arose. Involving peer research associates also contributed to the trauma-informed and decolonizing approach by offering a level of comfort and understanding, as well as empathizing with the experiences of FN-PWLE. This increased the level of trust between the participants and interviews which assisted with possibly gaining richer and more in-depth information from the participants. Finally, participants were provided with information regarding how to connect with wellness, mental health and Elder supports at the conclusion of the interview and all participants living with HIV were provided with an honorarium.

## Data analysis

The interviews were audio recorded, transcribed, edited, anonymized, and entered into NVivo (version 12.7) qualitative data analysis software. Each interview transcript was analysed using the following process: First, after each interview, the two interviewers populated a debriefing form. The form was used to summarize the interview and flag key ideas from the participants narrative. Second, the fully anonymized transcripts were analyzed by the wider study team by reading through each interview prior to meeting as a group (including the peer research associates, Elder and other team members) to discuss the unexpected revelations, new learnings or apparent themes each interview contained. Third, transcripts were manually coded using

inductive content analysis. This method was most suitable as it was used to analyze textual data by identifying and categorizing themes or patterns that emerged directly from the data without relying on pre-existing theories or predetermined coding frameworks [25]. Fourth, additional analysis included using NVivo to identify all the COVID-19 related sections of the transcripts (sentences and paragraphs) containing the words "COVID", "COVID-19" or "pandemic". Finally, two of the academic researchers on the team reviewed all codes and grouped them into themes based on emerging patterns. The analysis was then discussed with the peer research associates and the rest of the team to gain further insight and contribute to the analysis, thus leading to a coherent and congruent shared understanding of what the data is telling us. Findings relevant to the research question of this manuscript were extracted and prepared for reporting.

## Results

Twenty-five (25) participants were interviewed, including eleven HCPs and fourteen FN-PWLE. Seventeen participants lived in a large southern Manitoba urban center (Winnipeg, Manitoba), while eight lived in one of three smaller northern cities (Thompson, Flin Flon or The Pas).

The ages of the FN-PWLE ranged between 25–59 years old (median age 42.6 years), while the HCPs' ages ranged between 30–60 years (median 42.2 years). The median length of time since receiving an HIV diagnosis was 16.5 years (minimum two years, maximum 34 years). Ethnicity was identified as First Nation (Cree, Ojibwa, Dene, Oji-Cree, other) (n = 14 FN-PWLE; n = 4 (HCPs) and non-Indigenous (n = 0 FN-PWLE; n = 7 (HCPs). In total, 18/25 or 72% of the study participants self-identified as First Nation people.

Participants reported their gender (man, woman, non-binary) and sexual orientation (heterosexual, gay, lesbian, two-spirit, bisexual, asexual, queer, pansexual, other), but the numbers are repressed for this paper due to the small numbers in each category and confidentiality concerns.

Three key overarching themes and five subthemes emerged from throughout data analysis:

1. Disruptions to the health care (cascade of care) processes including:

   a. Disrupted relationships: how COVID-19 disrupted relationships and supportive care programs that were important to HIV care.

   b. Disrupted access to testing, in-person appointments and medications.

2. Intersectional stigma was compounded.

3. Creation of innovative ways to provide care, including:

   a. Enhanced use of virtual platforms for healthcare staff communication.

   b. Novel ways to access HIV medications in urban areas.

   c. Flexibility for accessing healthcare.

### Theme 1. Disruptions to the health care (cascade of care) processes

**1a. Disrupted relationships.**   According to the HCPs and FN-PWLE in the study, establishing and maintaining a relationship that involved trust and respect was an essential part of HIV health care. COVID-19 changed how HCPs and FN-PWLE interacted with each other and it decreased opportunities for nurturing the supportive relationships that existed between healthcare providers and clients.

*"I think what [healthcare] works well when a therapeutic relationship is built, and there is like rapport with the client. When it's a trusting relationship, it seems, the client is more ready to express needs and concerns"* (HCP).

The relationship between an HCP and a FN-PWLE relied on shared communication, trust, and respect. This relationship ensured that a FN-PWLE could be comfortable in vulnerable moments. The relational aspects of health care that were highly valued changed during the pandemic.

*"But you know what? I've not gotten used to it [COVID-19] but you know. . .. I put on my mask. I put on my face shield, I put on the gown. I can't tie the gowns in the back because of, you know, my size and because of my tremors. But there's a nurse over there that knows me. She ties the back when I can tie the front one, you know. I just do my thing and, you know, treat people with the utmost respect and compassion because we're all human beings, regardless of what we got or what we don't get or what we're getting or what we're going to get. You know, we're just all going down the road"* (FN-PWLE).

Before COVID-19, there were opportunities available for FN-PWLE to connect with HCPs and other FN-PWLE for educational opportunities and knowledge sharing. In-person programs and gatherings (food bank, educational sessions, social/cultural group meetings (i.e. Sisters of Fire, which is an Indigenous-led support group for women and two-spirited individuals living in Winnipeg)) were cancelled or significantly altered during the pandemic to strictly limit interpersonal contact. For months at a time, people were advised not to gather, not share food or have in-person conversations. The foodbank program at Nine Circles Community Health Care Centre in Winnipeg, was a social opportunity as well as means of obtaining groceries. Health centers were hubs where people gathered, attended appointments and educational programs, and meet to socialize.

*"And basically, these workshops where [pre] COVID, we were gathering with other like-minded individuals and you're able to sit together with other newly diagnosed people and other people that were just as scared as me and being able to get better together"* (FN-PWLE).

*"They had workshops there where for nice things for us. We had people from the university that are in training for massage therapy give us massages"* (FN-PWLE).

*"Nine Circles [Community Health Center] helps us. They have a food bank, and they give us the main things like bread, eggs, milk and stuff like that. There was meat sometimes and I think vegetables and everything to give us think the main food that we need. It used to be every two weeks, but now they moved it for every month because of COVID"* (FN-PWLE).

*"People walking in—that I would feed, or give a jacket to or tampons"* (HCP).

The study illustrated the importance of wrap-around care, providing non-medical care support and having touch points with the health system and health care provider team. Nine Circles Community Health Centre provides comprehensive community health services including HIV testing, treatment and wrap-around supports. The cancellation of services and supports limited people's ability to connect with others, socialize, and access amenities. For example, "*. . .because of COVID, we can't do outreach anymore*" (HCP). The restrictions combined with the changes in how health care was delivered by HCPs, altered or changed the relationship between HCPs and FN-PWLE during the pandemic. Years of dedicated energy, financial

resources, and planning to build and maintain relationship were paused by COVID-19 or needed to be re-configured to ensure people's safety.

**1b. Disrupted access to testing, in-person appointments and medications.** Three areas of the HIV care cascade were shown to be disrupted by the study: access to testing, in-person appointments and medications.

**Access to testing.** One of the essential healthcare processes that was disrupted due to the COVID-19 epidemic was HIV testing. Participants described that HIV testing rates went down during the COVID-19 lockdowns.

*"For sure, our testing numbers aren't what they should be compared to last year. Testing numbers were fourteen hundred and this year, it was six hundred. So, we're less than half of people coming in looking for routine testing"* (HCP).

The HCP attributed the decreased HIV/STBBI testing rates to the change in healthcare delivery processes, which were designed to limit potential COVID-19 transmission.

*"So, the flow of STBBI testing has changed for us. So, at the clinic, one of our testing sites, it's an automatic telephone appointment, and then they do the assessment over the phone. And then, if you need testing like urine and blood, then you can come in. But sometimes, it's scheduled a couple of days later for that. So, I think that is creating a barrier"* (HCP).

Testing opportunities were lost because of the move to telephone appointments.

*"There [was an] instance[s] where someone just wanted routine testing, and they couldn't show up. They couldn't attend their scheduled appointment to get the specimen collection. So, they were kind of, I guess, lost a testing opportunity because there was someone who wanted testing, and we just did the phone assessment. Whereas if they would have been there, we would have got the specimens…."* (HCP).

People were concerned about the lack of confidentiality at nursing stations in remote communities. Therefore, people seeking an HIV test might be reluctant to be tested at a nursing station, especially as they could not to travel to larger centers for testing during the COVID-19 lockdowns.

*"So, the people who rely on health care provision outside of the community have been cut off from that for a while"* (HCP). As a result of the more limited access the healthcare, it was speculated that HIV case numbers will increase after the lockdowns end. *"I think that we are going to find more cases [among people] who didn't go for testing while there was a shutdown"* (HCP).

**Access to in-person appointments.** The general pandemic restrictions combined with the changes in how healthcare was delivered by HCPs altered or changed the relationship between HCP and FN-PWLE during the pandemic. Healthcare providers replaced regular in-person appointments with telephone calls. Early in the pandemic an HCP noted that:

*"seventy-five percent [of appointments occurred] on the phone. Some people can't access a phone or don't enjoy seeking care over the phone. I mean, certainly, for a more complicated diagnosis like HIV, it would be less than ideal"* (HCP).

Participants in the study identified that changing to telephone appointments and using virtual platforms for meetings had unintended consequences. Telephone communication during the pandemic was identified as both a barrier and facilitator for healthcare. The average phone call was said to be "*six minutes*" (HCP). A quick phone call to renew a prescription and not having to "*in and see a doctor*" or wait made access easier, but HCPs were quick to point out that not all their clients had their own phones or even had access to telephones (HCP). The use of telephone appointments did not facilitate the fulsome doctor/client relationship that was afforded by a face-to-face meeting, and the inability to have a private conversation could compromise communication or increase the risk of miscommunication. "*And it's difficult, it's difficult to have somebody say taking care of your kids while you're trying to talk to the doctor or know in private*" (HCP).

However, HCPs in the north had to find workarounds to that they could communicate with their clients. "*Facebook was a way that [we communicated], and Facebook messenger was another way in which our program was accessing clients*" (HCP).

Manitoba Telehealth was already in before the pandemic to provide specialized HIV care, treatment and support for people living with HIV. The Telehealth program operates through the health authorities and has "sites" at specific locations in some First Nations communities and larger centres in various regions in Manitoba. (https://mbtelehealth.ca/locations/). However, the findings showed that some clients preferred to come to appointments rather than use Telehealth to ensure privacy (HCP). In addition, not everyone complied with the new procedures for having regular appointments over the telephone. In response to the question about any changes in the format of medical appointments, a participant responded that "*Oh no, no way. I told them when I first met Dr. XXX. . . . you. . . either see me [expletive] face-to-face or find me a new doctor*" (FN-PWLE).

Some FN-PWLE study participants who had computers or smartphones accepted the need for telephone appointments during the public health lockdowns. However, they felt that appointments with physicians and nurses would have been better/easier if they could see the HCP using virtual platforms.

> "*We have this we have these computers and tablets and phones now. So even if we could do like a Zoom meeting with the doctor, that would be a lot better than just on the phone because you only hear a voice. So, you can't really see them. You know, you can't really visit everybody*" (FN-PWLE).

One FN-PWLE suggested that when they had to talk to a nurse or doctor they were unfamiliar with, they felt very uncomfortable and preferred seeing a face rather than just hearing a voice. One FN-PWLE described how they missed the social interactions that surrounded their visit to the doctor's office. Telephone appointments lacked the personal interactions afforded by in-person clinic visits.

> "*I miss my doctor's office . . .. I talk to the nurse, and then I talk to the social worker. And then I talked to another nurse and then I talked to my doctor, so it's changed. I miss everybody in the office. My daughter came to my appointments, and they used to always have crayons and paper for her to doodle on and stuff like that*" (FN-PWLE).

**Access to medications.**   During the COVID-19 lockdown, there were delays for people getting their medications in northern communities.

> "*They weren't even letting the delivery people [in] that were dropping off the medications*" (HCP).

*"I can't tell you which communities and how many of them, but I know that that was a concern for the pharmacists, obviously, was that we were hearing from patients that their medications weren't being received. [In] the early days of when the communities were locked down, there definitely was, in some cases, a gap. But, you know, the people that are adherent obviously get extremely concerned when there's a gap in their medications arriving"* (HCP).

## Theme 2. Intersectional stigma was compounded

For FN-PWLE, factors contributing to intersectional stigma multiplied during the COVID-19 lockdowns. During the early months of the pandemic before vaccinations were available, the fear of contracting the disease was of significant concern for those who were immune-compromised and those with pre-existing mental health conditions. Participants recounted how COVID-19 resulted in additional mental health stress and situations where they were re-traumatized, thereby increasing feelings of being alone.

*"With COVID, it's impacted me greatly, tremendously. Now I have anxiety with, you know, not only in PTSD anxiety, I have depression now because I don't have the ability to see people. And because I'm HIV [positive], I'm one of those high-risk people that can get it, and it can kill us"* (FN-PWLE).

The lack of attention on the impacts of COVID-19 on people living with HIV left them feeling unsupported, further stigmatized, and deprioritized.

*"And that's another thing [about having HIV] that put us [persons with HIV or who self-identify as First Nation] on the backburner without getting any support and help"* (FN-PWLE).

One participant highlighted that in some rural areas, there was a lack of acknowledgement that diseases like HIV and COVID-19 even exist.

*"I just I find it frustrating here because when you live in [community name], people have this idea that HIV doesn't exist here, COVID doesn't exist here. . ."* (FN-PWLE).

One participant saw similarities in the stigma and fear of contracting HIV during the 1980s to the current fear and uncertainties around COVID-19. The participants living with HIV saw some irony in the public health orders related to COVID-19, which in their opinion, treated everyone as equals, as if everyone had an infection and should be avoided.

*"In this current climate, we should treat everyone as if they're HIV positive. We should treat everyone as if they're COVID positive. You know, COVID is the new HIV"* (FN-PWLE).

## Theme 3. Creation of innovative ways to provide care

There were several lessons learned from study participants about health systems improvements. Several innovations emerged during the pandemic crisis, where necessity spawned invention. These were categorized into three sub-themes that suggest beneficial systems change for people living with HIV.

**3a. Enhanced and flexible use of virtual platforms for healthcare staff communication.** HCPs noted that because of COVID-19, computer software was made available to

accommodate the need to connect with FN-PWLE, and efforts to connect with clients by phone increased.

> *"The gift of COVID, it has been like, our health authority fast-tracked Microsoft teams in for us. So as tricky as Zoom can be, we actually can call them [FN-PWLE] and Zoom-in, and they don't always have internet, and there's always some other problems, but we're getting better at connecting by phone"* (HCP).

COVID-19 seemed to increase innovation in healthcare delivery. The study HCP participants supported this innovation by highlighting a need for alternative methods for connecting with FN-PWLE.

> *"Diversifying the ways, we provide virtual care, and care closer to the community, can serve to make that [healthcare]better as well. It allowed us to have virtual care that is more flexible using different platforms"* (HCP).

However, at the same time, many First Nations people in Manitoba did not have access to the internet and technology, especially in northern communities or due to poverty. During lockdowns, while some could connect using virtual platforms, many FN-PWLE lacked access to phones or computers and were, therefore, more isolated.

> *"I feel sorry for all the people that can't access Wi-Fi. They're missing out. They're going stir crazy at home? that are feeling suicidal, that can't get on a [expletive] phone or a Wi-Fi or a Zoom conference to help them better themselves because they don't have access to the Internet"* (FN-PWLE).

> *"Not all of us can afford computers. You think all of us have computers? I'm lucky that I have Wi-Fi. And, you know, this COVID thing is really impacted a lot of us because not all of us have Wi-Fi"* (FN-PWLE).

**3b. Novel ways to access HIV medications in urban areas.**   Access to medications was not impacted by COVID-19 if there was a previously established system between the FN-PWLE, the doctor and a pharmacy.

> *"Sometimes the challenge is that when we have a positive diagnosis, we connect with HIV Manitoba, and those meds are prescribed through the primary care provider but in conjunction, or in consultation, with the team and that's been a pretty smooth process."* (HCP).

In some respects, the increased dependence on telephone appointments improved the system for accessing medications.

> *"If someone needed refills instead of going into an appointment, it was easier for them to just do that appointment over the phone"* (HCP).

> *"I haven't heard anything that there's been any issues; if anything, I think it's a little bit better, more organized. . . . They're trying to prevent people kind of just coming into the nursing stations. It might be actually easier for clients to get their medication depending on the individual community and what the process has been"* (HCP).

However, it was also reported that the cancellation of regular physician appointments meant that prescription refills needed to be arranged by the FN-PWLE and as a result, there were complications getting refills.

FN-PWLE appreciated that during COVID-19 lockdowns, they were able to have more medication on hand.

*". . . we were able to take more medication home than usual, so that way we weren't having to go out every day or every couple of weeks. So, we had one or two-months' supply of medication"* (FN-PWLE).

## Discussion

The COVID-19 pandemic created radical reductions in the ability of the healthcare system to respond to the needs of the public in Manitoba, in the rest of Canada and throughout the world [26, 27]. In addition, the social restrictions required to control the pandemic created layers of challenges. The negative impacts of the pandemic disproportionately affected vulnerable members of society, including but not limited to those with pre-existing health concerns, those who were economically and socially disadvantaged, older adults, and racial and other stigmatized groups [2, 28–30]. This study illustrated that because of the COVID-19 pandemic, there were effects on the health system for First Nations people living with HIV, including; disrupted relationships between First Nations people living with HIV and healthcare providers; disrupted access to testing, in-person appointments and medication; intersectional stigma was compounded; and innovative solutions for the health system.

### Disrupted relationships between people living with HIV and healthcare providers

One of the most critical study findings was the impact of COVID-19 on relationships between healthcare providers and their clients. Cancelling in-person programs and services during the COVID-19 lockdown impacted important relationships between HCPs and FN-PWLE. In addition, participants described how face-to-face check-ins in the past proved to be important for the overall care experience. The connections afforded by in-person contact could not be replicated by using the telephone for medical appointments. FN-PWLE thought medical appointments on the phone would have been more comfortable with a more secure feeling if they had had a visual component rather than just audio. However, the innovation of the increased use of virtual platforms during the pandemic demonstrated how connections and relationships could be maintained without the need for in-person contact. FN-PWLE appreciated the opportunity to see people using Zoom or Teams software. During the COVID-19 lockdown, FN-PWLE in this study reported that they valued receiving care from HCPs with whom they had an established relationship.

### Disrupted access to testing, in-person appointments, and medications

Participants in this study described how the pandemic caused disruptions to the HIV cascade of care processes (Theme 1) through reduced availability and funding to programs and services. Reduced access to health services during the pandemic decreased mental wellness and increased stress, as reported by participants (Theme 2). This same finding regarding the effects of COVID-19 on First Nations people's mental health was also noted in an Albert First Nations community [31]. Some basic services, such as food banks for people with structural disadvantages were disrupted during the COVID-19 epidemic. Some study participants appreciated the

continuation of some services during the COVID-19 lockdowns (e.g. the foodbank) yet recognized the need to modify the service for health safety. For FN-PWLE consistency, and targeted interventions for overall wellness and healthcare provision is an aspect of care that are recognized as a priority for healthcare retention to ensure optimal outcomes [32, 33].

## Intersectional stigma was compounded

The COVID-19 pandemic resulted in several societal fears and resulting stigmas. For communities that already experience prejudice and stigma, the pandemic compounded the effects in an intersectional manner. FN-PWLE in this study reported intersectional stigmatization, which is supported by other studies reporting changes in the healthcare delivery that enhanced stigmatization [32]. Other racialized groups also reported intersectional stigma during the pandemic. Stigma related to having the COVID-19 virus and being of Asian descent drew attention in the media and public health [34].

Lessons learned from HIV epidemics helped Public Health organizations and others create messaging to reduce COVID-19 stigma. This included using first-person language, speaking positively, engaging social influencers, telling stories, and using images of local people who have or had COVID-19 to address myths, rumours and stereotypes [5, 35]. Experiences with HIV stigmatization led to the recommendation that instead of criminalizing people who breach COVID-19 Public Health policies, communities should be empowered and strengthened to support people to protect themselves regarding others' health [35, 36].

## Innovative solutions for health systems

At the time of the study, Manitoba had yet to adopt virtual physician appointments during the pandemic, unlike other provinces such as British Columbia. Although there are potential problems in the use of virtual care, such as issues of security, privacy issues and quality of care, it is an option for individuals who seek face-to-face interactions with their HCPs [37]. As people became familiar with virtual meetings during the pandemic, FN-PWLE in this study indicated that they would feel less isolated if they could have had access to virtual visits where they would have the ability to see and communicate with the care provider.

## Recommendations

1. Strategies to reduce the risk of losing these relationships for any reason (service cutbacks, pandemics) should be a priority for health systems planners. Relationships between HCPs and FN-PWLE are critical for retaining people in care, and action should be taken to sustain and promote the critical HCP/ FN-PWLE relationships. For example, wrap-around care and attention to social determinants of health (food, income, housing, etc.) are critical aspects of relational health services.

2. Lessons learned from the COVID-19 pandemic for providing education and for helping to address intersectional stigma may be beneficial and applicable to addressing these same issues as related to HIV. For example, one of the many ways the First Nation Health and Social Secretariat of Manitoba (an organization dedicated to First Nation health) responded to the COVID-19 pandemic was to provide supportive, culturally appropriate, and immediately relevant information on a dedicated website [38]. A webpage dedicated to raising awareness of HIV, sharing knowledge for newly diagnosed First Nation people, facilitating learning about HIV testing and treatment options, and providing linkages to supportive organizations (housing, food security, social) could help address some intersectional stigma.

3. Innovations such as virtual care and innovative testing options (self-testing kits similar to COVID-19 rapid antigen kits, dry blood spot testing, etc.) can be beneficial. They should be explored further if in-person appointments are impossible for various reasons, including geography, health orders, convenience, etc. A comprehensive system analysis and multi-stakeholder consultation for both urban and remote and rural HIV services on how best to implement these options should be undertaken by health system planners.

## Study limitations

The HCPs were interviewed within the first six months of the Public Health orders regarding COVID-19 lockdowns in Manitoba. In contrast, the FN-PWLE were interviewed after the Public Health orders had been in place for several months. As a result, the FN-PWLE had more time to observe, experience and process the impacts than the HCPs.

Conducting interviews using a virtual platform or on the telephone presented some limitations. One potential participant was not interviewed due to their lack of access to the internet. We were aware that some people could not enroll in the study due to a lack of access to phones or computers, but we could not track how many. Economic discrimination (poverty) and the lack of affordable technology contributed to inequitable access to our study and wellness supports during COVID-19.

The use of Zoom had some limitations with the FN-PWLE group. For some FN-PWLE, the Zoom platform was new, and their access was on phones or tablets rather than computers. As a result, some FN-PWLE either chose or could not access the video feature in Zoom, creating a feeling of isolation or talking into a void. Alternatively, interviews with HCPs were easier to perform via Zoom than in-person. HCPs had computer equipment and software available to them, they had experience using Zoom, they were comfortable using the platform, and both the HCPs and researchers appreciated that there was no travel time for the meeting.

The researchers were aware of the power imbalance created for the FN-PWLE during the qualitative interview process. Despite consenting to be interviewed and explaining the purpose of the study, FN-PWLE may still have felt vulnerable. Including peer research associates with lived experience helped address this imbalance and gave lived experience voice and understanding to the data analysis.

## Conclusions

For Manitoba First Nation people living with HIV, the COVID-19 pandemic exaggerated pre-existing health system barriers. As well, the pandemic brought to like some health systems facilitators such as the attention to relationships and social supports, particularly for First Nation people and those who are structurally disadvantaged and needing more wrap-around care to address social determinants of health.

First Nation persons living with HIV who participated in the study emphasized the importance of the relationship between the HCP to care continuity. In addition, the pandemic highlighted and, in some cases, exaggerated existing issues and gaps in the HIV healthcare system (such as decreased access to testing) and in society (such as intersectional stigma) for First Nations people in Manitoba. Alternatively, the pandemic made us aware of the need to innovate how we care for those in need, using methods that support but do not erode the trusting relationships built through in-person interaction.

The study also illustrated the importance of qualitative research about HIV healthcare delivery to inform systems change. The voices of both the First Nation people and the healthcare providers that work with them were heard and they identified how HIV healthcare could

change to provide better services to meet the needs of people living with HIV. This study illustrates the importance and value of using First Nation decolonizing methods in research. Including First Nation people on our study team enhanced the interview and analysis process.

Lessons learned from the COVID-19 pandemic should be considered for improvements to the health system's HIV cascade of care for FN-PWLE as well as for other health system improvements for First Nations people. Working together with First Nations organizations and communities, further studies are needed to strategize and apply the recommendations and innovations, to change healthcare and people's lives.

## Acknowledgments

We gratefully acknowledge the study participants for sharing their insights.

## Author Contributions

**Conceptualization:** Linda Larcombe, Laurie Ringaert, Albert McLeod, Pamela Orr.

**Data curation:** Linda Larcombe, Gayle Restall.

**Formal analysis:** Linda Larcombe, Laurie Ringaert, Gayle Restall, Albert McLeod, Elizabeth Hydesmith, Ann Favel, Melissa Morris.

**Funding acquisition:** Linda Larcombe, Laurie Ringaert, Gayle Restall, Albert McLeod, Michael Payne, Yoav Keynan, Kelly MacDonald, Pamela Orr.

**Investigation:** Linda Larcombe, Laurie Ringaert, Gayle Restall, Albert McLeod, Elizabeth Hydesmith, Ann Favel, Melissa Morris, Kelly MacDonald, Pamela Orr.

**Methodology:** Linda Larcombe, Laurie Ringaert, Gayle Restall, Albert McLeod, Elizabeth Hydesmith, Rusty Souleymanov, Yoav Keynan, Matthew Singer, Pamela Orr.

**Project administration:** Linda Larcombe, Laurie Ringaert, Gayle Restall, Michael Payne, Matthew Singer.

**Resources:** Linda Larcombe, Laurie Ringaert, Rusty Souleymanov.

**Supervision:** Linda Larcombe, Laurie Ringaert, Gayle Restall, Elizabeth Hydesmith, Matthew Singer.

**Validation:** Linda Larcombe, Laurie Ringaert, Gayle Restall, Albert McLeod, Elizabeth Hydesmith, Ann Favel, Melissa Morris, Matthew Singer.

**Visualization:** Linda Larcombe, Laurie Ringaert, Gayle Restall.

**Writing – original draft:** Linda Larcombe, Laurie Ringaert, Albert McLeod, Pamela Orr.

**Writing – review & editing:** Linda Larcombe, Laurie Ringaert, Gayle Restall, Albert McLeod, Elizabeth Hydesmith, Ann Favel, Melissa Morris, Michael Payne, Rusty Souleymanov, Yoav Keynan, Kelly MacDonald, Matthew Singer, Jared Star, Pamela Orr.

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
