## [Decision Letter · Decision Letter 0]

10 Apr 2023

PONE-D-22-28673“Because of Covid…”: The impacts of Covid-19 on First Nation people accessing the HIV cascade of care in Manitoba, Canada.PLOS ONE

Dear Dr. Larcombe,

Thank you for submitting your manuscript to PLOS ONE. After careful consideration, we feel that it has merit but does not fully meet PLOS ONE’s publication criteria as it currently stands. Therefore, we invite you to submit a revised version of the manuscript that addresses the points raised during the review process.

We look forward to receiving your revised manuscript.

Kind regards,

Miracle Ayomikun Adesina, BPT

Academic Editor

PLOS ONE

“LL, LR, AM, GR, PO, KM, YK, MS, received the Canadian Institutes of Health Research (CIHR) Operating Grant (Grant Number CBA-164022)”

4. Thank you for stating the following in the Acknowledgments/ Funding Section of your manuscript:

“Canadian Institutes of Health Research (CIHR) Mapping the Journey: Developing Culturally Appropriate, Geographically-Responsive HIV Care for Northern Manitoba First Nation People. CIHR Operating Grant (Grant Number CBA-164022”

“LL, LR, AM, GR, PO, KM, YK, MS, received the Canadian Institutes of Health Research (CIHR) Operating Grant (Grant Number CBA-164022)”

5.Thank you for stating the following in your Competing Interests section: 

“The authors declare not conflicts of interest.”

7. Your ethics statement should only appear in the Methods section of your manuscript. If your ethics statement is written in any section besides the Methods, please delete it from any other section.

8. PLOS requires an ORCID iD for the corresponding author in Editorial Manager on papers submitted after December 6th, 2016. Please ensure that you have an ORCID iD and that it is validated in Editorial Manager. To do this, go to ‘Update my Information’ (in the upper left-hand corner of the main menu), and click on the Fetch/Validate link next to the ORCID field. This will take you to the ORCID site and allow you to create a new iD or authenticate a pre-existing iD in Editorial Manager. Please see the following video for instructions on linking an ORCID iD to your Editorial Manager account: https://www.youtube.com/watch?v=_xcclfuvtxQ.

Additional Editor Comments:

General comments

1. There are some grammatical and sentence structure errors that needs attention

2. It is not essential to name study team members in the manuscript. Moreover, the named co-principal investigator is not indicated as an author on the manuscript

3. All quotations should be presented in the results section

4. Discussion of the results in relation to previous studies and implications on health systems and policy could enhanced

5. The study limitation could be summarised

Specific comments

1. As a qualitative study the authors stated using questionnaires as the data collection tool in the abstract. This should be corrected appropriately

2. The second paragraph of the introduction on page 5 contains methods and results. This section should focus on the problem and gap(s) in research that this study seeks to fill

3. The “understanding the context” section could be summarised and added to the methods as the study setting

4. The following statement imply an implicit bias in the analysis of the data "The peer research associates brought to the interviews and analysis their knowledge, experience and insights that are unique to people with lived experience”. As part of the research team their knowledge and experience should not influenced the analysis

5. Provide the ethical approval numbers

6. Provide months in which data collection began and ended

7. Information on the type of demographic collected has been repeated on page 11

8. The authors indicated that both inductive and deductive analysis techniques were used in the study. The authors should be clear on how and why both analysis techniques were used.

9. The authors indicated that data on gender and sexual orientation cannot be reported due to the small number and confidential issues. All the reported data has been anonymised so I don’t think there is any risk of breaching confidentiality. Furthermore, differences of opinions on gender and sexual orientation could be essential for designing interventions

10. Some of the quotes can be shortened by focusing on the main information. Also, some quotes could be grouped together under one narrative for better coherence. For instance, the delays in getting appointment is associated with delays in receiving medications. These two points could be merged. Each quotation should also be presented in separate paragraph with more identifiers of the participant such as age and/or sex

11. There is no need to itemize the themes before reporting the detailed results. This leads to avoidable repetitions

12. Consider rephrasing "Disruption to overall well-being....” to "Decline of overall well-being..."

13. On page 16 "... ability to have a private conversation cold compromised communication” should be corrected to “... the inability to…"

14. On page 16, the following sentence does not connect with the previous statements and the subsequent quote does not match the narrative: "However, HCPs in the north had to find work arounds so that they could communicate with their clients. “Facebook was a way that Facebook messenger was another way in which our program was accessing clients” (HCP)." Further clarification is required

15. The sentence "If an individual tried to make an appointment at a northern clinic." On page 19 is seems out place

16. On page 20, in the sentence “The lack of attention on the impacts of Covid-19 on people who were living with HIV left those with lived experience feeling unsupported, further stigmatized and deprioritized” needs to be appropriately phrased. It currently presents people living with HIV and those with lives experience as different groups of people.

17. The sub-themes under theme 3 are very similar particularly themes 3A and 3C. I suggest those two themes be combined

18. The second and third paragraph of the discussion could be merged since they present similar information.

19. On page 26 the authors indicated that “The Covid-19 epidemic began at the time we were planning to start our interviews and thus our planned study process was significantly disrupted.” The authors need to restructure this sentence appropriately since it implies the impact of the Covid-19 epidemic is not the main objective of the study.

20. The second paragraph of the conclusion is redundant. The conclusion should be more directed to the implication of the findings to improving healthcare delivery for marginalised people and PWLE

Reviewers' comments:

Reviewer's Responses to Questions

**Comments to the Author**

1. Is the manuscript technically sound, and do the data support the conclusions?

Reviewer #1: Yes

Reviewer #2: Yes

2. Has the statistical analysis been performed appropriately and rigorously? 

Reviewer #1: N/A

Reviewer #2: N/A

3. Have the authors made all data underlying the findings in their manuscript fully available?

Reviewer #1: Yes

Reviewer #2: Yes

4. Is the manuscript presented in an intelligible fashion and written in standard English?

Reviewer #1: Yes

Reviewer #2: No

5. Review Comments to the Author

Reviewer #1: The manuscript is well written--the introduction section explains the research clearly and in a lucid manner, making it easy to understand. The methods section is well detailed and the results section is elaborate.

Reviewer #2: General comments

1. There are some grammatical and sentence structure errors that needs attention

2. It is not essential to name study team members in the manuscript. Moreover, the named co-principal investigator is not indicated as an author on the manuscript

3. All quotations should be presented in the results section

4. Discussion of the results in relation to previous studies and implications on health systems and policy could enhanced

5. The study limitation could be summarised

Specific comments

1. As a qualitative study the authors stated using questionnaires as the data collection tool in the abstract. This should be corrected appropriately

2. The second paragraph of the introduction on page 5 contains methods and results. This section should focus on the problem and gap(s) in research that this study seeks to fill

3. The “understanding the context” section could be summarised and added to the methods as the study setting

4. The following statement imply an implicit bias in the analysis of the data "The peer research associates brought to the interviews and analysis their knowledge, experience and insights that are unique to people with lived experience”. As part of the research team their knowledge and experience should not influenced the analysis

5. Provide the ethical approval numbers

6. Provide months in which data collection began and ended

7. Information on the type of demographic collected has been repeated on page 11

8. The authors indicated that both inductive and deductive analysis techniques were used in the study. The authors should be clear on how and why both analysis techniques were used.

9. The authors indicated that data on gender and sexual orientation cannot be reported due to the small number and confidential issues. All the reported data has been anonymised so I don’t think there is any risk of breaching confidentiality. Furthermore, differences of opinions on gender and sexual orientation could be essential for designing interventions

10. Some of the quotes can be shortened by focusing on the main information. Also, some quotes could be grouped together under one narrative for better coherence. For instance, the delays in getting appointment is associated with delays in receiving medications. These two points could be merged. Each quotation should also be presented in separate paragraph with more identifiers of the participant such as age and/or sex

11. There is no need to itemize the themes before reporting the detailed results. This leads to avoidable repetitions

12. Consider rephrasing "Disruption to overall well-being....” to "Decline of overall well-being..."

13. On page 16 "... ability to have a private conversation cold compromised communication” should be corrected to “... the inability to…"

14. On page 16, the following sentence does not connect with the previous statements and the subsequent quote does not match the narrative: "However, HCPs in the north had to find work arounds so that they could communicate with their clients. “Facebook was a way that Facebook messenger was another way in which our program was accessing clients” (HCP)." Further clarification is required

15. The sentence "If an individual tried to make an appointment at a northern clinic." On page 19 is seems out place

16. On page 20, in the sentence “The lack of attention on the impacts of Covid-19 on people who were living with HIV left those with lived experience feeling unsupported, further stigmatized and deprioritized” needs to be appropriately phrased. It currently presents people living with HIV and those with lives experience as different groups of people.

17. The sub-themes under theme 3 are very similar particularly themes 3A and 3C. I suggest those two themes be combined

18. The second and third paragraph of the discussion could be merged since they present similar information.

19. On page 26 the authors indicated that “The Covid-19 epidemic began at the time we were planning to start our interviews and thus our planned study process was significantly disrupted.” The authors need to restructure this sentence appropriately since it implies the impact of the Covid-19 epidemic is not the main objective of the study.

20. The second paragraph of the conclusion is redundant. The conclusion should be more directed to the implication of the findings to improving healthcare delivery for marginalised people and PWLE

6. PLOS authors have the option to publish the peer review history of their article (what does this mean?). If published, this will include your full peer review and any attached files.

Reviewer #1: No

Reviewer #2: No

---

## [Author Response · Author response to Decision Letter 0]

8 Jun 2023

Re: PONE-D-22-28673 “Because of Covid…”: The impacts of Covid-19 on First Nation people accessing the HIV cascade of care in Manitoba, Canada.

In response to the Journal requirements

1. We have ensured that our manuscript meets PLOS ONE’s style requirements.

2. We have clarified in the manuscript and on the submission form that consent from the participants was obtained in a written form.

3. In response to the financial disclosure request for additional information we provide the following statement

• Funding:

o Canadian Institutes of Health Research (CIHR) Mapping the Journey: Developing Culturally Appropriate, Geographically-Responsive HIV Care for Northern Manitoba First Nation People. CIHR Operating Grant (Grant Number CBA-164022). The funders had no role in study design, data collection and analysis, the decision to publish, or the preparation of the manuscript.

4. We have removed information about the funders from the acknowledgement section. It now reads “We gratefully acknowledge the study participants for sharing their insights.” 

5. We have declared that "The authors have declared that no competing interests exist."

6. The qualitative data used to for the creation of this manuscript are included in the document therefore additional data storage is not necessary. 

7. We have moved the ethics statement into the Methods section of the manuscript.

8. I have updated the ORCID iD for the corresponding author. 

Responses to the Editor’s comments

1. There are some grammatical and sentence structure errors that needs attention

• We have edited the manuscript to correct the grammatical and sentence structure errors

2. It is not essential to name study team members in the manuscript. Moreover, the named co-principal investigator is not indicated as an author on the manuscript

• We have removed the name of the study team members from the manuscript.

• We have reviewed the list of authors and ensured that it is complete.

3. All quotations should be presented in the results section

• We have moved all the quotations to the results sections.

4. Discussion of the results in relation to previous studies and implications on health systems and policy could enhanced

• The discussion section has been rewritten to include how this study relates to previous studies we have presented recommendations for health systems and policies. 

5. The study limitation could be summarised

• We have summarized the study limitations.

Responses to the specific comments:

1. As a qualitative study the authors stated using questionnaires as the data collection tool in the abstract. This should be corrected appropriately

• We have changed the abstract to indicate that we in fact used an interview guide and not a questionnaire. 

2. The second paragraph of the introduction on page 5 contains methods and results. This section should focus on the problem and gap(s) in research that this study seeks to fill

• We have removed the sentences that contain methods and results on this page so that the introduction is focused on the problem and gaps in research.

3. The “understanding the context” section could be summarised and added to the methods as the study setting

• We have moved the “understanding the context” section into the methods and removed it as a sub-section.

4. The following statement imply an implicit bias in the analysis of the data "The peer research associates brought to the interviews and analysis their knowledge, experience and insights that are unique to people with lived experience”. As part of the research team their knowledge and experience should not influenced the analysis

• We have augmented this section of the manuscript with references that address the merits and indeed, the best practice for having peer researchers involved in all aspects of the research. The research team’s breadth of knowledge was enhanced by the peers’ experience and wisdom that is informed by the intersectionality of living with HIV and being First Nations. In the Study Design section we provide a summary about the importance of having peer research associates and references that support this approach. 

5. Provide the ethical approval numbers

• The University of Manitoba Health Ethics Review Board numbers for this project have been included on page 8.

6. Provide months in which data collection began and ended

• We have included the months in which the data collection began and ended.

7. Information on the type of demographic collected has been repeated on page 11

• On page 10 we documented the demographic information collected from First Nation people with lived experience and on page 11 from Health Care Providers. Some of the data collected was the same for both groups (age, gender, sexual orientation, current residence) but other information was specific to each group. For people with lived experience for example we also asked about the 

8. The authors indicated that both inductive and deductive analysis techniques were used in the study. The authors should be clear on how and why both analysis techniques were used.

9. The authors indicated that data on gender and sexual orientation cannot be reported due to the small number and confidential issues. All the reported data has been anonymised so I don’t think there is any risk of breaching confidentiality. Furthermore, differences of opinions on gender and sexual orientation could be essential for designing interventions

• We have reviewed our participant numbers and found that in some categories we had between 1 to 3 participants. The First Nation people with lived HIV experience in Manitoba who agreed to participate in this study form a small group and are known to each other, and to groups involved in activism, research and community work. We have considered the sensitivity of the data and the potential harm the could result from identification of individuals among this vulnerable group. We respectfully maintain that there is a risk and that the numbers should remain repressed.

10. Some of the quotes can be shortened by focusing on the main information. Also, some quotes could be grouped together under one narrative for better coherence. For instance, the delays in getting appointment is associated with delays in receiving medications. These two points could be merged. Each quotation should also be presented in separate paragraph with more identifiers of the participant such as age and/or sex

• Given our small sample size more information about the individuals who provided the quotes might put the participants at risk of being identified. In addition, we did not analyze our data by sex/gender or age so we are not convinced that providing this information with the quotes would add to the narratives. 

11. There is no need to itemize the themes before reporting the detailed results. This leads to avoidable repetitions

• The itemizing of the themes has been removed to avoid repetition.

12. Consider rephrasing "Disruption to overall well-being....” to "Decline of overall well-being..."

• We have changed this Theme to “Intersectional Stigma” thereby removing any commentary about the participants responses and to focus the issues they raise related to stigma. 

13. On page 16 "... ability to have a private conversation cold compromised communication” should be corrected to “... the inability to…"

• “ability” has been corrected and changed to “inability”

14. On page 16, the following sentence does not connect with the previous statements and the subsequent quote does not match the narrative: "However, HCPs in the north had to find work arounds so that they could communicate with their clients. “Facebook was a way that Facebook messenger was another way in which our program was accessing clients” (HCP)." Further clarification is required

• We have included additional language in the quote to clarify that HCPs used both Facebook and Facebook messenger to communicate with clients. “Facebook was a way that [we communicated] and Facebook messenger was another way in which our program was accessing clients” (HCP).

15. The sentence "If an individual tried to make an appointment at a northern clinic." On page 19 is seems out place

• We have included additional language to clarify this sentence. “If an individual tried to make an appointment at a northern clinic they were required to disclose personal health information to people other than their physician.” 

16. On page 20, in the sentence “The lack of attention on the impacts of Covid-19 on people who were living with HIV left those with lived experience feeling unsupported, further stigmatized and deprioritized” needs to be appropriately phrased. It currently presents people living with HIV and those with lives experience as different groups of people.

• We have edited this sentence to make it clearer and it now reads “The lack of attention on the impacts of Covid-19 for people who were living with HIV left them feeling unsupported, further stigmatized, and deprioritized.” 

17. The sub-themes under theme 3 are very similar particularly themes 3A and 3C. I suggest those two themes be combined

• We agree and we have combined sub-themes 3A and 3C.

18. The second and third paragraph of the discussion could be merged since they present similar information.

• We have edited the discussion section to make it more concise.

19. On page 26 the authors indicated that “The Covid-19 epidemic began at the time we were planning to start our interviews and thus our planned study process was significantly disrupted.” The authors need to restructure this sentence appropriately since it implies the impact of the Covid-19 epidemic is not the main objective of the study. 

• We have edited the study limitations section and clarified the interview process in the methods section.

20. The second paragraph of the conclusion is redundant. The conclusion should be more directed to the implication of the findings to improving healthcare delivery for marginalised people and PWLE.

• We have heavily edited the conclusions and removed the redundancies. 

Re: PONE-D-22-28673 “Because of Covid…”: The impacts of Covid-19 on First Nation people accessing the HIV cascade of care in Manitoba, Canada.

Dear Editors,

We appreciate the careful review provided by the reviewers and their suggestions for improving the content of this manuscript.

In response to the Journal requirements

1. We have ensured that our manuscript meets PLOS ONE’s style requirements.

2. We have clarified in the manuscript and on the submission form that consent from the participants was obtained in a written form.

3. In response to the financial disclosure request for additional information we provide the following statement

• Funding:

o Canadian Institutes of Health Research (CIHR) Mapping the Journey: Developing Culturally Appropriate, Geographically-Responsive HIV Care for Northern Manitoba First Nation People. CIHR Operating Grant (Grant Number CBA-164022). The funders had no role in study design, data collection and analysis, the decision to publish, or the preparation of the manuscript.

4. We have removed information about the funders from the acknowledgement section. It now reads “We gratefully acknowledge the study participants for sharing their insights.” 

5. We have declared that "The authors have declared that no competing interests exist."

6. The qualitative data used to for the creation of this manuscript are included in the document therefore additional data storage is not necessary. 

7. We have moved the ethics statement into the Methods section of the manuscript.

8. I have updated the ORCID iD for the corresponding author. 

Responses to the Editor’s comments

1. There are some grammatical and sentence structure errors that needs attention

• We have edited the manuscript to correct the grammatical and sentence structure errors

2. It is not essential to name study team members in the manuscript. Moreover, the named co-principal investigator is not indicated as an author on the manuscript

• We have removed the name of the study team members from the manuscript.

• We have reviewed the list of authors and ensured that it is complete.

3. All quotations should be presented in the results section

• We have moved all the quotations to the results sections.

4. Discussion of the results in relation to previous studies and implications on health systems and policy could enhanced

• The discussion section has been rewritten to include how this study relates to previous studies we have presented recommendations for health systems and policies. 

5. The study limitation could be summarised

• We have summarized the study limitations.

Responses to the specific comments:

1. As a qualitative study the authors stated using questionnaires as the data collection tool in the abstract. This should be corrected appropriately

• We have changed the abstract to indicate that we in fact used an interview guide and not a questionnaire. 

2. The second paragraph of the introduction on page 5 contains methods and results. This section should focus on the problem and gap(s) in research that this study seeks to fill

• We have removed the sentences that contain methods and results on this page so that the introduction is focused on the problem and gaps in research.

3. The “understanding the context” section could be summarised and added to the methods as the study setting

• We have moved the “understanding the context” section into the methods and removed it as a sub-section.

4. The following statement imply an implicit bias in the analysis of the data "The peer research associates brought to the interviews and analysis their knowledge, experience and insights that are unique to people with lived experience”. As part of the research team their knowledge and experience should not influenced the analysis

• We have augmented this section of the manuscript with references that address the merits and indeed, the best practice for having peer researchers involved in all aspects of the research. The research team’s breadth of knowledge was enhanced by the peers’ experience and wisdom that is informed by the intersectionality of living with HIV and being First Nations. In the Study Design section we provide a summary about the importance of having peer research associates and references that support this approach. 

5. Provide the ethical approval numbers

• The University of Manitoba Health Ethics Review Board numbers for this project have been included on page 8.

6. Provide months in which data collection began and ended

• We have included the months in which the data collection began and ended.

7. Information on the type of demographic collected has been repeated on page 11

• On page 10 we documented the demographic information collected from First Nation people with lived experience and on page 11 from Health Care Providers. Some of the data collected was the same for both groups (age, gender, sexual orientation, current residence) but other information was specific to each group. For people with lived experience for example we also asked about the 

8. The authors indicated that both inductive and deductive analysis techniques were used in the study. The authors should be clear on how and why both analysis techniques were used.

9. The authors indicated that data on gender and sexual orientation cannot be reported due to the small number and confidential issues. All the reported data has been anonymised so I don’t think there is any risk of breaching confidentiality. Furthermore, differences of opinions on gender and sexual orientation could be essential for designing interventions

• We have reviewed our participant numbers and found that in some categories we had between 1 to 3 participants. The First Nation people with lived HIV experience in Manitoba who agreed to participate in this study form a small group and are known to each other, and to groups involved in activism, research and community work. We have considered the sensitivity of the data and the potential harm the could result from identification of individuals among this vulnerable group. We respectfully maintain that there is a risk and that the numbers should remain repressed.

10. Some of the quotes can be shortened by focusing on the main information. Also, some quotes could be grouped together under one narrative for better coherence. For instance, the delays in getting appointment is associated with delays in receiving medications. These two points could be merged. Each quotation should also be presented in separate paragraph with more identifiers of the participant such as age and/or sex

• Given our small sample size more information about the individuals who provided the quotes might put the participants at risk of being identified. In addition, we did not analyze our data by sex/gender or age so we are not convinced that providing this information with the quotes would add to the narratives. 

11. There is no need to itemize the themes before reporting the detailed results. This leads to avoidable repetitions

• The itemizing of the themes has been removed to avoid repetition.

12. Consider rephrasing "Disruption to overall well-being....” to "Decline of overall well-being..."

• We have changed this Theme to “Intersectional Stigma” thereby removing any commentary about the participants responses and to focus the issues they raise related to stigma. 

13. On page 16 "... ability to have a private conversation cold compromised communication” should be corrected to “... the inability to…"

• “ability” has been corrected and changed to “inability”

14. On page 16, the following sentence does not connect with the previous statements and the subsequent quote does not match the narrative: "However, HCPs in the north had to find work arounds so that they could communicate with their clients. “Facebook was a way that Facebook messenger was another way in which our program was accessing clients” (HCP)." Further clarification is required

• We have included additional language in the quote to clarify that HCPs used both Facebook and Facebook messenger to communicate with clients. “Facebook was a way that [we communicated] and Facebook messenger was another way in which our program was accessing clients” (HCP).

15. The sentence "If an individual tried to make an appointment at a northern clinic." On page 19 is seems out place

• We have included additional language to clarify this sentence. “If an individual tried to make an appointment at a northern clinic they were required to disclose personal health information to people other than their physician.” 

16. On page 20, in the sentence “The lack of attention on the impacts of Covid-19 on people who were living with HIV left those with lived experience feeling unsupported, further stigmatized and deprioritized” needs to be appropriately phrased. It currently presents people living with HIV and those with lives experience as different groups of people.

• We have edited this sentence to make it clearer and it now reads “The lack of attention on the impacts of Covid-19 for people who were living with HIV left them feeling unsupported, further stigmatized, and deprioritized.” 

17. The sub-themes under theme 3 are very similar particularly themes 3A and 3C. I suggest those two themes be combined

• We agree and we have combined sub-themes 3A and 3C.

18. The second and third paragraph of the discussion could be merged since they present similar information.

• We have edited the discussion section to make it more concise.

19. On page 26 the authors indicated that “The Covid-19 epidemic began at the time we were planning to start our interviews and thus our planned study process was significantly disrupted.” The authors need to restructure this sentence appropriately since it implies the impact of the Covid-19 epidemic is not the main objective of the study. 

• We have edited the study limitations section and clarified the interview process in the methods section.

20. The second paragraph of the conclusion is redundant. The conclusion should be more directed to the implication of the findings to improving healthcare delivery for marginalised people and FN-PWLE.

• We have heavily edited the conclusions and removed the redundancies.

---

## [Editor Report · Decision Letter 1]

10 Jul 2023

“Because of Covid…”: The impacts of Covid-19 on First Nation people accessing the HIV cascade of care in Manitoba, Canada.

PONE-D-22-28673R1

Dear Dr. Larcombe,

We’re pleased to inform you that your manuscript has been judged scientifically suitable for publication and will be formally accepted for publication once it meets all outstanding technical requirements.

Kind regards,

Miracle Ayomikun Adesina, BPT

Academic Editor

PLOS ONE
---

## [Editor Report · Acceptance letter]

25 Jul 2023

PONE-D-22-28673R1 

“Because of COVID…”: The impacts of COVID-19 on First Nation people accessing the HIV cascade of care in Manitoba, Canada. 

Dear Dr. Larcombe:

I'm pleased to inform you that your manuscript has been deemed suitable for publication in PLOS ONE. Congratulations! Your manuscript is now with our production department. 

Kind regards, 

on behalf of

Dr. Miracle Ayomikun Adesina 

Academic Editor

PLOS ONE